# Unveiling Air Pollution in Crimean Mountain Rivers: Analysis of Sentinel-5 Satellite Images Using Google Earth Engine (GEE)

**Vladimir Tabunschik [1], Roman Gorbunov [1,*] and Tatiana Gorbunova [1,2]**

1   A.O. Kovalevsky Institute of Biology of the Southern Seas of RAS, 299011 Sevastopol, Russia; tabunshchyk@ibss-ras.ru (V.T.); gorbunovatyu@ibss-ras.ru (T.G.)
2   Institute of Environmental Engineering, Department of Subtropical and Tropical Ecology, Peoples' Friendship University of Russia (RUDN University), 117198 Moscow, Russia
*   Correspondence: gorbunov@ibss-ras.ru

**Abstract:** This article presents an assessment of atmospheric pollutant concentrations based on state-of-the-art geoinformation research methods that utilize Sentinel-5 satellite imagery, the cloud computing platform Google Earth Engine (GEE), and ArcGIS 10.8 software. The spatial distributions of some pollutants (nitrogen dioxide, sulfur dioxide, formaldehyde, carbon monoxide, methane) in the atmosphere are analyzed on the example of the basins of the Zapadnyy Bulganak, Alma, Kacha, Belbek, and Chernaya rivers on the north-western slope of the Crimean Mountains. The concentrations of the average annual and average monthly values of pollutants for each catchment area are compared. The GEE (Google Earth Engine) platform is used for extracting annual and monthly average rasters of pollutant substances, while ArcGIS is utilized for enhanced data visualization and in-depth analytical processing. Background concentrations of pollutants within protected natural areas are calculated. By comparing the spatial and temporal distribution of pollutant values with the background concentrations within these protected areas, a complex index of atmospheric pollution is constructed. The spatial and temporal variability of nitrogen dioxide ($NO_2$) concentrations has been thoroughly examined. Based on the regression analysis (R > 0.85), the field of values of the total amount of emissions (which are analyzed for only six points in the study area and in the surrounding areas) was restored on the basis of the spatial and temporal heterogeneity of the field of distribution of nitrogen dioxide values ($NO_2$). Since air pollution can have negative consequences, both for human health and for the ecosystem as a whole, this study is of great importance for assessing the ecological situation within the river basins of the north-western slope of the Crimean Mountains. This work also contributes to a general understanding of the problem of gas emissions, whose study is becoming increasingly relevant. The aim of this research is to assess the potential application of Sentinel-5 satellite imagery for air quality assessment and pollution analysis within the river basins of the north-western slopes of the Crimean Mountains. The significance of this study lies in the innovative use of Sentinel-5 satellite imagery to investigate air pollution in extensive regions where a regular network of observation points is lacking.

**Keywords:** river; river basin; GIS; pollution; monitoring; Sentinel-5

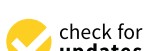



## 1. Introduction

Whether considered at local, regional, or global levels, atmospheric air pollution is an extremely urgent problem [1,2] due to its effects on human health [3,4] and the environment [5,6]. Among the main global causes of air pollution are emissions from automobile and aviation transport [7], emissions from industrial enterprises [8], emissions from mining [9], the consumption of large quantities of fossil fuels [10], and associated emissions from agricultural activities [11].

There are various methods used in the study of the problem of atmospheric air pollution. Among the leading approaches to the analysis of atmospheric air pollution

are those involving the use of monitoring systems, such as ground-based air quality sensors and satellite instruments. Such systems can provide detailed information about the concentration of various pollutants in the atmosphere, as well as their sources and distribution patterns. However, limitations affecting their joint use are primarily due to the role of ground-based monitoring systems, which impose a limitation in terms of the discrete number of spatiotemporal observations.

Due to the close integration of field and remote research methods, the use of computer models to study the behavior of pollutants in the atmosphere has increased in recent years [12], along with studies into the consequences to human health and organisms of exposure to various pollutants [13], as well as those considering the influence of various natural [14], social [15], and economic [16] factors in atmospheric air pollution.

A number of space satellites have been used to monitor the state of the atmosphere in recent years [17–19]. One of the most modern of these, providing a wide coverage of spatiotemporal data, is the Sentinel-5 Precursor satellite mission [20]. The Sentinel-5 Precursor is a satellite vehicle launched by the European Space Agency in October 2017 as part of the Copernicus Earth observation program. The purpose of the mission is to conduct global atmospheric measurements related to air quality, climate change, and monitoring of the ozone layer. The Sentinel-5 Precursor is equipped with the Tropomi instrument (TROPOspheric Monitoring Instrument), a state-of-the-art spectrometer for measuring the concentration of a number of gases in the Earth's atmosphere, including nitrogen dioxide, sulfur dioxide, ozone, carbon monoxide, methane, formaldehyde, and aerosols.

The theoretical foundations of the use and technical characteristics of the ESA/EU Copernicus Sentinel-5 Precursor satellite are described in the works [21,22]. Since its 2018 launch, data obtained by the ESA/EU Copernicus Sentinel-5 Precursor satellite has been widely used. In [23] Vîrghileanu et al., nitrogen dioxide ($NO_2$) pollution monitoring was carried out over Europe during the coronavirus pandemic outbreak using Sentinel-5 Precursor satellite images; in Zheng et al., spatial variation of $NO_2$ and its impact factors in China were investigated [24]; Kaplan et al. [25] used Sentinel-5 Precursor satellite images to study the prevalence of nitrogen dioxide over Turkey; Kaplan and Avdan [26] investigated the distribution of CO and $NO_2$ in Northern Macedonia using Sentinel-5 Precursor satellite images; Schneider et al. [27] studied the spatiotemporal distribution of $NO_2$ over Norway. Magro et al. [28] analyzed atmospheric trends of CO and $CH_4$ from extreme wildfires in Portugal; Theys et al. reported on global monitoring of volcanic $SO_2$ degassing [29]; Safarianzengir et al. [30] conducted monitoring and analysis of spatial and temporal zoning of air pollution (carbon monoxide) for health management in Iran; Mazlan et al. [31] studied the relationship between the reduction of gas emissions and quarantine conditions imposed due to the COVID-19 pandemic in Peninsular Malaysia; Alvarado et al. [32] revealed the long-range transport of glyoxal and formaldehyde observed from the Copernicus Sentinel-5 Precursor satellite during the 2018 Canadian wildfires.

Thus, the ESA/EU Copernicus Sentinel-5 Precursor satellite data can be said to support a wide and constantly increasing research geography and scope of coverage. ESA/EU Copernicus Sentinel-5 Precursor satellite data on the content of chemicals in the atmosphere, which could not be collected in such quantities by means of ground field expeditions, lends itself to remote analysis. Moreover, some regions lack an effective network for monitoring observations of the state of the atmosphere and the content of various impurities and gases in the atmosphere. In order to help solve this problem, the ESA/EU Copernicus Sentinel-5 Precursor satellite data becomes invaluable.

One of these regions is the Crimean Peninsula. In general, for an area comprising around 26,000 $km^2$, there is an extremely poorly developed network of observation points for sources of atmospheric pollution. The study of Crimean atmospheric pollution can be said to have a sporadic character, as well as often being performed using various incompatible methods, which significantly reduces the comparability of data. However, there are a number separate works on the study of atmospheric pollution for the Crimean Peninsula. Pollution is often studied in the context of administrative territorial units.

Tabunshchyk et al. [33] studied emissions from stationary pollution sources in the Republic of Crimea over the period 2013–2018. There are also works considering general issues involved in the study of atmospheric air pollution over the Crimean Peninsula [34,35]. Nevertheless, most of the relevant studies investigate particular aspects of pollution without considering the entire territory of the peninsula holistically [36–41]. The content of chemical elements in atmospheric precipitation is investigated in the works [42,43]. Nekhoroshkov et al. [36] studied the distribution of heavy metals and other elements of atmospheric origin in the Crimean Mountains using the moss biomonitoring method. Varenik [37] analyzed air pollution $PM_{2.5}$ and $PM_{10}$ with elemental carbon in Sevastopol; Lapchenko and Zvyagintsev [38] studied the characteristics of atmospheric gases in the Karadag Nature Reserve in Crimea; Zvyagintsev et al. [39] investigated air pollution over the Crimean Peninsula during the hot summer of 2010. Much attention is paid to the analysis of pollution in large settlements—Sevastopol [37,44], Simferopol [45,46], Yalta [41,47], etc. As emphasized in [48–50], an increase in atmospheric emissions affects the health of the Crimean population, increasing the number of respiratory pathologies.

There are even fewer studies of atmospheric air pollution within the basins of the Zapadnyy Bulganak, Alma, Kacha, Belbek, and Chernaya rivers; here, the majority of attention is paid instead to the study of pollution of river waters [51–54], as well as marine waters [55,56], into which rivers carry out a large quantity of pollutants. There are only a few studies of air pollution within the river basins under consideration. For example, Kashirina et al. [57] investigated the conditions of surface atmospheric air in the south-western Crimea according to lichen-indication data. Nekrich [40] provides a map of the impact of agricultural burdens on the environment of the Crimean Peninsula in 2021. This study concludes that the greatest impacts on the ecosystems of the river basins of the north-western slope of the Crimean occurs in the Zapadnyy Bulganak River, as well as in the lower and middle reaches of the Alma, Kacha, Belbek, and Chernaya rivers.

The main objectives of the present work are: to show the spatiotemporal variability of pollutant concentration fields within the river basins of the north-western slope of the Crimean Mountains calculated from Sentinel-5 Precursor satellite images; to establish a link between the concentration of pollutants from satellite images and the results of monitoring atmospheric pollution (for individual settlements); on the basis of satellite imagery data, to reconstruct the concentration field of pollutant emissions. To achieve the aforementioned objectives, the following research materials were utilized: satellite imagery data from the Sentinel-5 spacecraft, statistical data on emissions monitoring within inhabited areas located within river basins or adjacent to river basin boundaries, and vector geospatial data depicting river basin boundaries and boundaries of protected natural areas. This holds particular significance for regions encompassing substantial study areas where conventional data collection methods face limitations. Leveraging the utilization of Sentinel-5 satellite imagery empowers the monitoring and assessment of air quality across expansive territories, thereby bestowing a more comprehensive grasp of the prevailing environmental milieu.

## 2. Materials and Methods

### 2.1. Study Area

The territory of the basins of the Zapadnyy Bulganak, Alma, Kacha, Belbek, and Chernaya rivers is located in the Crimean Peninsula (Figure 1). The area of the studied territory comprises approximately 2299 km$^2$ [58].

### 2.2. Methods

European Space Agency datasets obtained from the Copernicus Sentinel-5 Precursor satellite mission were used as initial data on the content of various pollutants in the atmosphere. The Sentinel-5 mission consists of a high-resolution spectrometer system that operates within the ultraviolet to shortwave infrared range, utilizing seven distinct spectral bands: UV-1 (270–300 nm), UV-2 (300–370 nm), VIS (370–500 nm), NIR-1 (685–710 nm),

NIR-2 (745–773 nm), SWIR-1 (1590–1675 nm), and SWIR-3 (2305–2385 nm). The instrument will be hosted on the MetOp-SG A satellite. To simplify the procedure for obtaining data from the Copernicus Sentinel-5 Precursor satellite (simplification of data processing of netCDF files), the Google Earth Engine (GEE) was used to calculate monthly and annual average concentrations of nitrogen dioxide, sulfur dioxide, formaldehyde, methane, and carbon monoxide, as well as the value of the aerosol index. Comprising a cloud-based computing platform for analyzing and processing large-scale geospatial data, GEE provides a powerful and flexible environment for working with a wide range of remote-sensing, satellite, and other geospatial datasets, including Sentinel-5 Precursor data. One of the key advantages of using GEE for geospatial analysis consists in its ability to efficiently process and analyze huge amounts of data without the need for expensive computing equipment or software. GEE also offers a collaborative environment for sharing data, code, and analysis results with others.

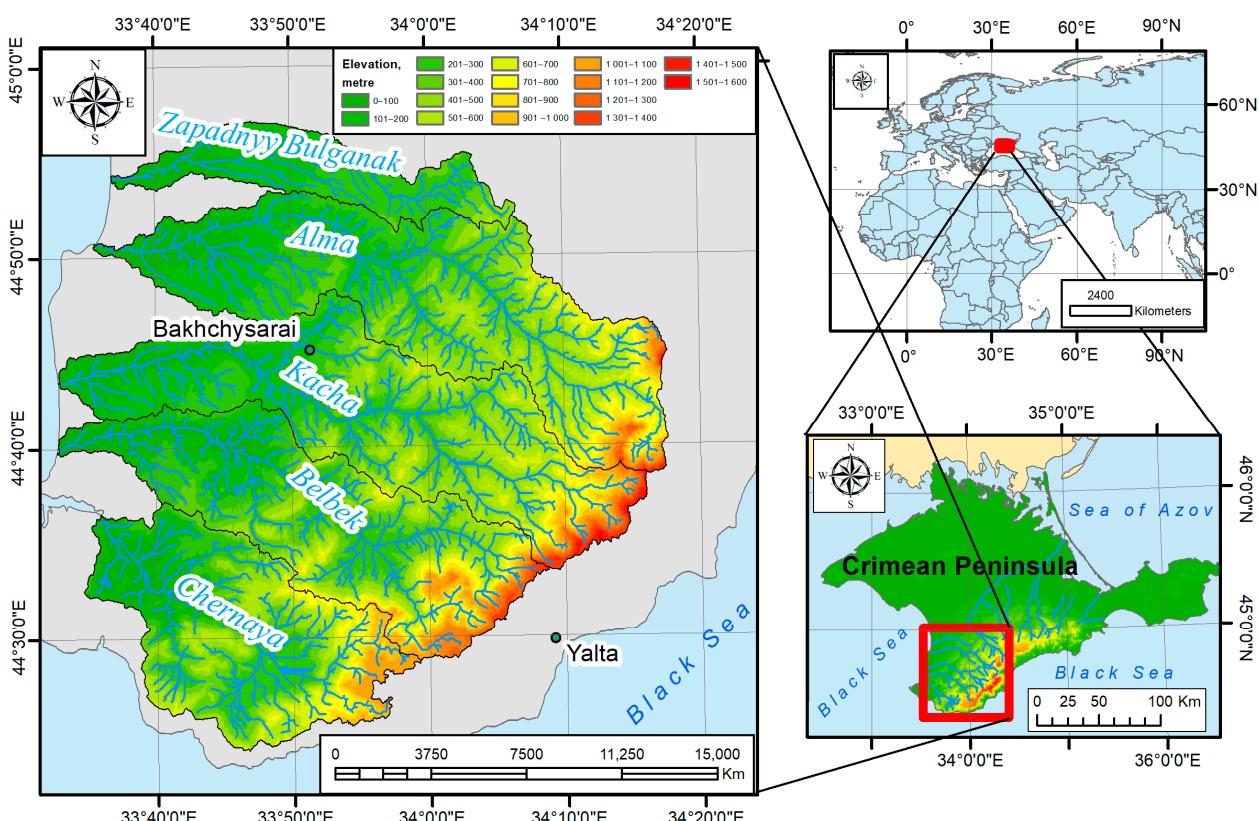

**Figure 1.** Geographical location of the research area [58].

The "Sentinel-5P L3" collection was used to estimate the concentration of pollutants (for example, for nitrogen dioxide—ee.ImageCollection (COPERNICUS/S5P/OFFL/L3_NO2)). For the collection, various methods of filtering (obtaining average annual and monthly values) and data processing (cropping along the boundaries of the studied area) and further analysis of the resulting raster values of pollutants were used. The obtained data was subsequently stored using Google Drive. Visualization of geographical maps was performed using the ArcGIS 10.8 software package, which provides more opportunities for visually analyzing the received data.

A modified indicator of the complex index of atmospheric pollution (CIAP) used to account for the combined effects of all pollutants is calculated by the formula [59]:

$$\text{CIAP} = \left( \sum_{i=1}^{n} \frac{qi}{Fi} \right) c, \tag{1}$$

where *i* is a pollutant;

$q_i$—average annual pollutant concentration;

$F_i$—corresponding average daily maximum allowable concentration;

*c*—constant taking values ($c = 1.7$ (hazard class I of the pollutant substance); $c = 1.3$ (hazard class II of the pollutant substance); $c = 1.0$ (hazard class III of the pollutant substance); $c = 0.9$ (hazard class IV of the pollutant substance));

*n*—number of impurities.

Data on the total amount of emissions within the study area and nearby settlements as a result of field monitoring were obtained from the official statistical directories of the Republic of Crimea [60,61] and the city of Sevastopol [62,63]. The total emission data are presented in Table 1.

Regression analysis of the relationship of emissions in cities based on the results of the study of satellite images and monitoring data was carried out using RStudio and Microsoft Excel software.

The research scheme can be represented graphically (Figure 2).

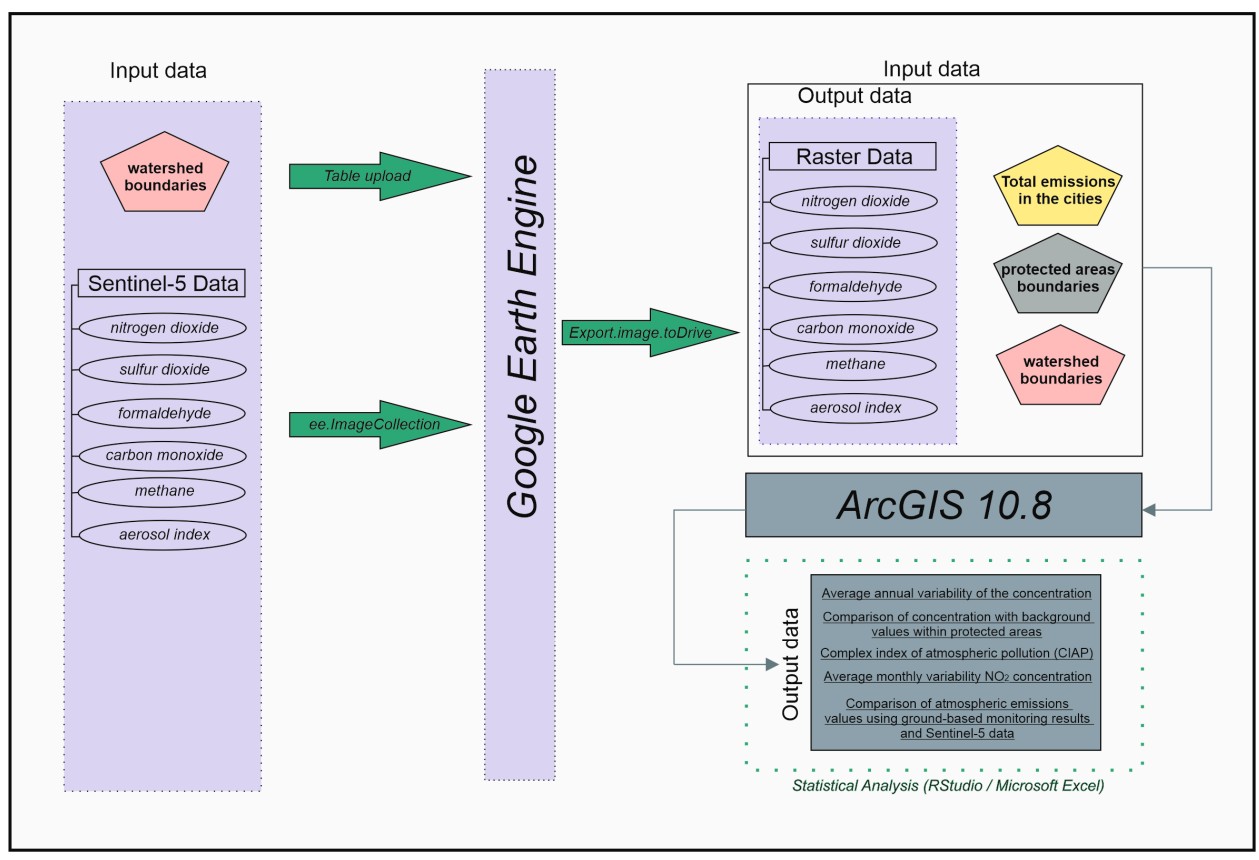

**Figure 2.** Scheme of the study of air pollution in river basins using Google Earth Engine and ArcGIS.

**Table 1.** Total emissions (thousand tons) in the cities.

| City | Year | | | |
|---|---|---|---|---|
| | **2018** | **2019** | **2020** | **2021** |
| Simferopol | 4.229 | 5.986 | 7.092 | 7.811 |
| Sevastopol | 3.034 | 5.511 | 6.882 | 6.665 |
| Bakhchisaray | 1.732 | 2.258 | 2.468 | 2.52 |
| Yalta | 0.314 | 0.491 | 0.623 | 0.61 |
| Saki | 0.331 | 0.775 | 0.961 | 2.408 |
| Alushta | 0.268 | 0.414 | 0.395 | 0.395 |

## 3. Results

### 3.1. Average Annual Variability of the Concentration of Chemicals in the Atmosphere

The spatial field distribution of the substances under consideration in the atmosphere for 2018, 2019, 2020, 2021, and 2022 within the river basins of the north-western slope of the Crimean Mountains is shown in Figures 3–8 and in Table 2.

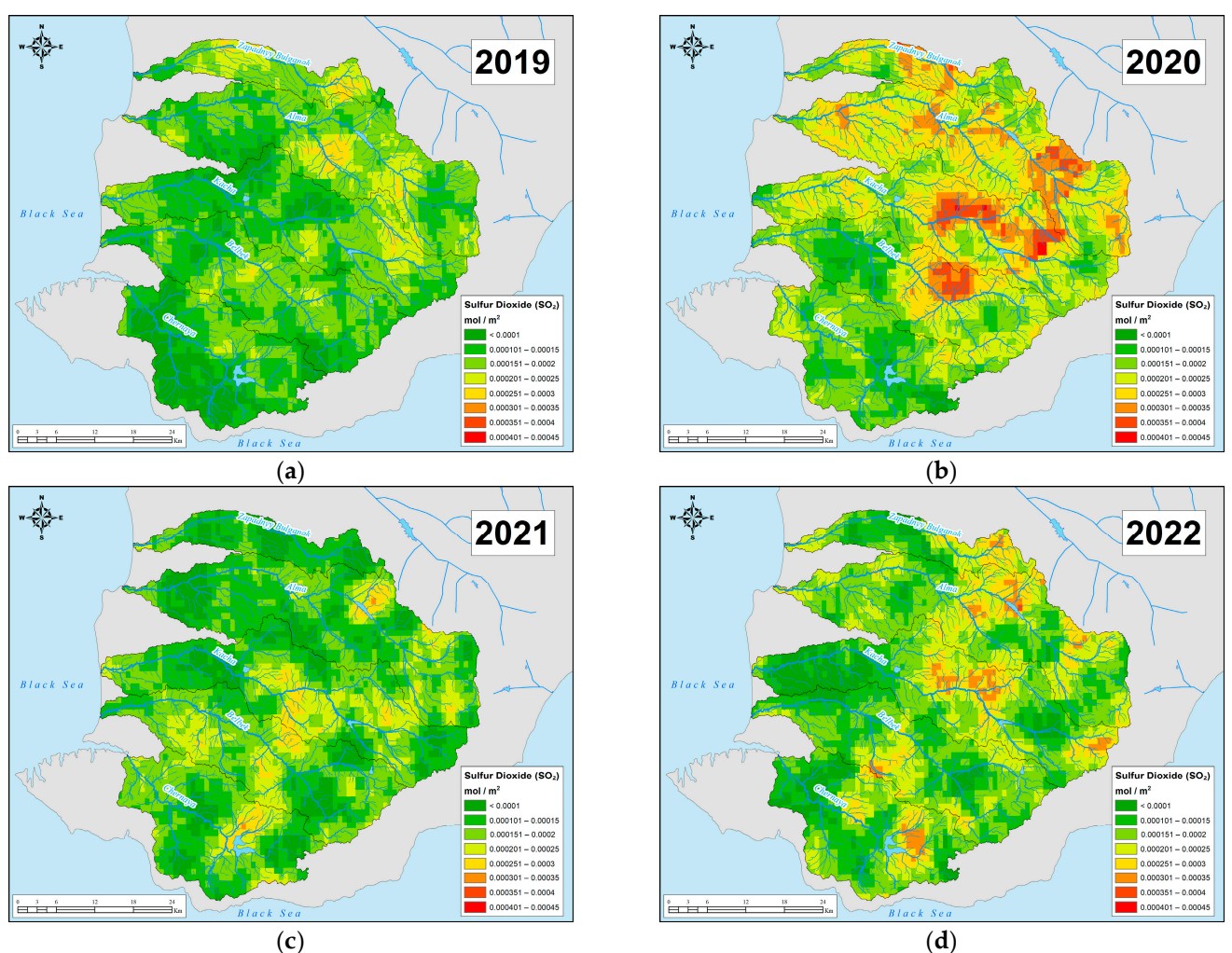

**Figure 3.** Distribution of sulfur dioxide (SO₂) content: (**a**) 2019; (**b**) 2020; (**c**) 2021; (**d**) 2022.

**Table 2.** Average annual atmospheric concentrations within the basins of the Zapadnyy Bulganak, Alma, Kacha, Belbek, and Chernaya rivers.

| Index. Average Value | Year | River Basins | | | | |
|---|---|---|---|---|---|---|
| | | Zapadnyy Bulganak | Alma | Kacha | Belbek | Chernaya |
| Aerosol Index | 2018 | −0.89 | −0.96 | −0.94 | −0.97 | −0.96 |
| | 2019 | −1.07 | −1.12 | −1.13 | −1.12 | −1.15 |
| | 2020 | −1.35 | −1.42 | −1.40 | −1.41 | −1.43 |
| | 2021 | −0.88 | −0.91 | −0.91 | −0.92 | −0.91 |
| | 2022 | −0.31 | −0.38 | −0.37 | −0.38 | −0.37 |
| Carbon Monoxide—CO (mol/m²) | 2018 | 0.031 | 0.030 | 0.030 | 0.030 | 0.031 |
| | 2019 | 0.033 | 0.032 | 0.031 | 0.031 | 0.032 |
| | 2020 | 0.033 | 0.032 | 0.032 | 0.032 | 0.033 |
| | 2021 | 0.035 | 0.034 | 0.033 | 0.033 | 0.034 |
| | 2022 | 0.031 | 0.030 | 0.030 | 0.030 | 0.030 |

**Table 2.** *Cont.*

| Index. Average Value | Year | River Basins | | | | |
|---|---|---|---|---|---|---|
| | | Zapadnyy Bulganak | Alma | Kacha | Belbek | Chernaya |
| Formaldehyde—HCHO $(mol/m^2)$ | 2018 | 0.000048 | 0.000043 | 0.000041 | 0.000039 | 0.000041 |
| | 2019 | 0.000099 | 0.000098 | 0.000101 | 0.000105 | 0.000099 |
| | 2020 | 0.000086 | 0.000086 | 0.000089 | 0.000088 | 0.000092 |
| | 2021 | 0.000082 | 0.000078 | 0.000084 | 0.000081 | 0.000089 |
| | 2022 | 0.000081 | 0.000084 | 0.000083 | 0.000086 | 0.000089 |
| Nitrogen Dioxide—$NO_2$ $(mol/m^2)$ | 2018 | 0.000027 | 0.000023 | 0.000022 | 0.000021 | 0.000023 |
| | 2019 | 0.000025 | 0.000022 | 0.000022 | 0.000021 | 0.000023 |
| | 2020 | 0.000025 | 0.000022 | 0.000021 | 0.000021 | 0.000022 |
| | 2021 | 0.000025 | 0.000022 | 0.000021 | 0.000020 | 0.000021 |
| | 2022 | 0.000025 | 0.000023 | 0.000022 | 0.000021 | 0.000022 |
| Sulfur Dioxide—$SO_2$ $(mol/m^2)$ | 2019 | 0.00019 | 0.00017 | 0.00015 | 0.00015 | 0.00012 |
| | 2020 | 0.00024 | 0.00025 | 0.00025 | 0.00020 | 0.00016 |
| | 2021 | 0.00012 | 0.00015 | 0.00016 | 0.00017 | 0.00016 |
| | 2022 | 0.00018 | 0.00020 | 0.00018 | 0.00017 | 0.00017 |

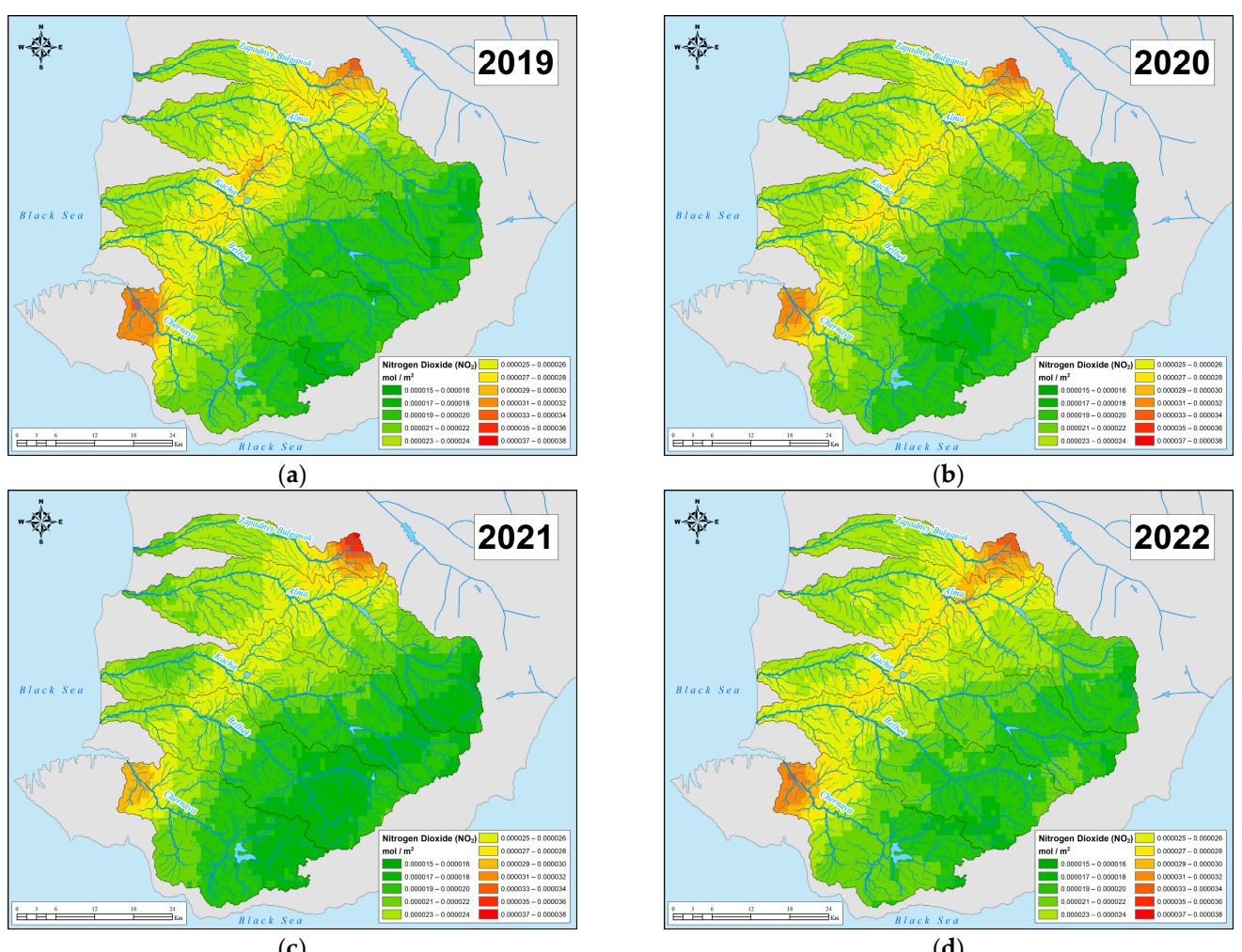

**Figure 4.** Distribution of nitrogen dioxide ($NO_2$) content: (**a**) 2019; (**b**) 2020; (**c**) 2021; (**d**) 2022.

Figure 3 shows the distribution of the average annual values of sulfur dioxide ($SO_2$) within the river basins of the north-western slope of the Crimean Mountains for the period 2019–2022. As can be seen from Figure 3, the average annual values of the sulfur dioxide field have a complex distribution. The lowest sulfur dioxide concentrations in the atmosphere are characteristic of the lower reaches of the Zapadnyy Bulganak, Alma, Kacha,

Belbek, and Chernaya rivers, while the highest concentrations correspond to the central parts of the river basins under consideration, as well as the upper part of the Zapadnyy Bulganak River basin.

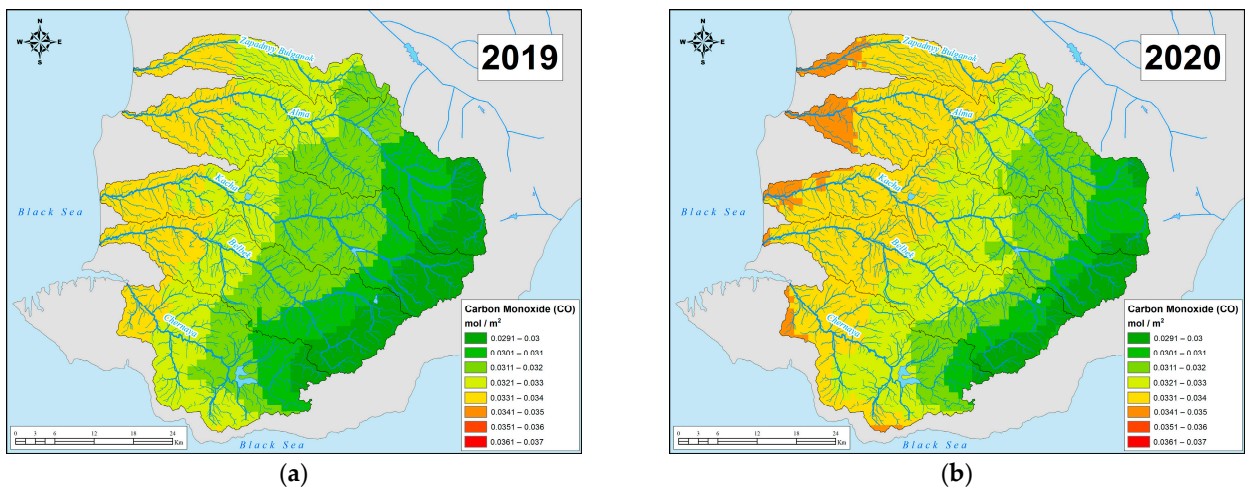

**Figure 5.** Distribution of formaldehyde (HCHO) content: (**a**) 2019; (**b**) 2020; (**c**) 2021; (**d**) 2022.

**Figure 6.** *Cont.*

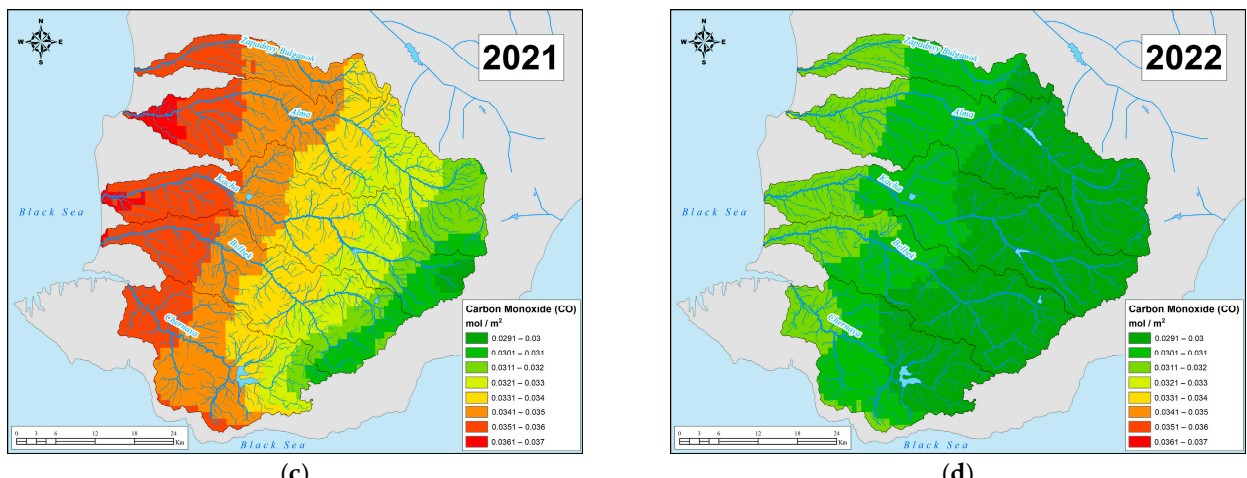

**Figure 6.** Distribution of carbon monoxide (CO) content: (**a**) 2019; (**b**) 2020; (**c**) 2021; (**d**) 2022.

Figure 4 shows the variability of the average annual values of nitrogen dioxide concentration within the river basins of the north-western slope of the Crimean Mountains. Figure 4 clearly shows that the maximum concentrations of nitrogen dioxide are typical over large cities—Sevastopol, Simferopol, Bakhchysarai—as well as along the path of major highways connecting these settlements. The lowest concentrations of nitrogen dioxide are characteristic of the area occupying the upper reaches of the Alma, Kacha, Belbek, and Chernaya river basins, where the minimum values of anthropogenic impact on ecosystems are observed.

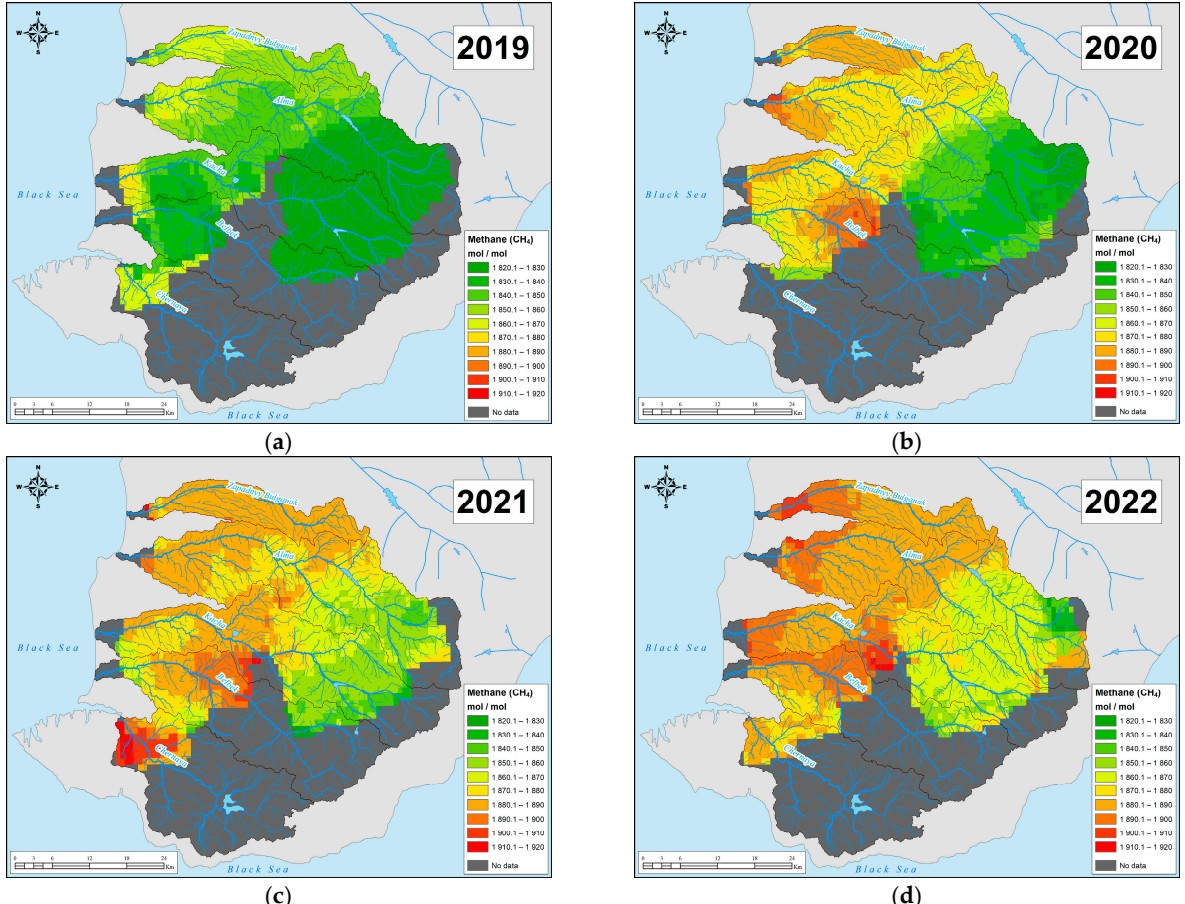

**Figure 7.** Distribution of methane ($CH_4$) content: (**a**) 2019; (**b**) 2020; (**c**) 2021; (**d**) 2022.

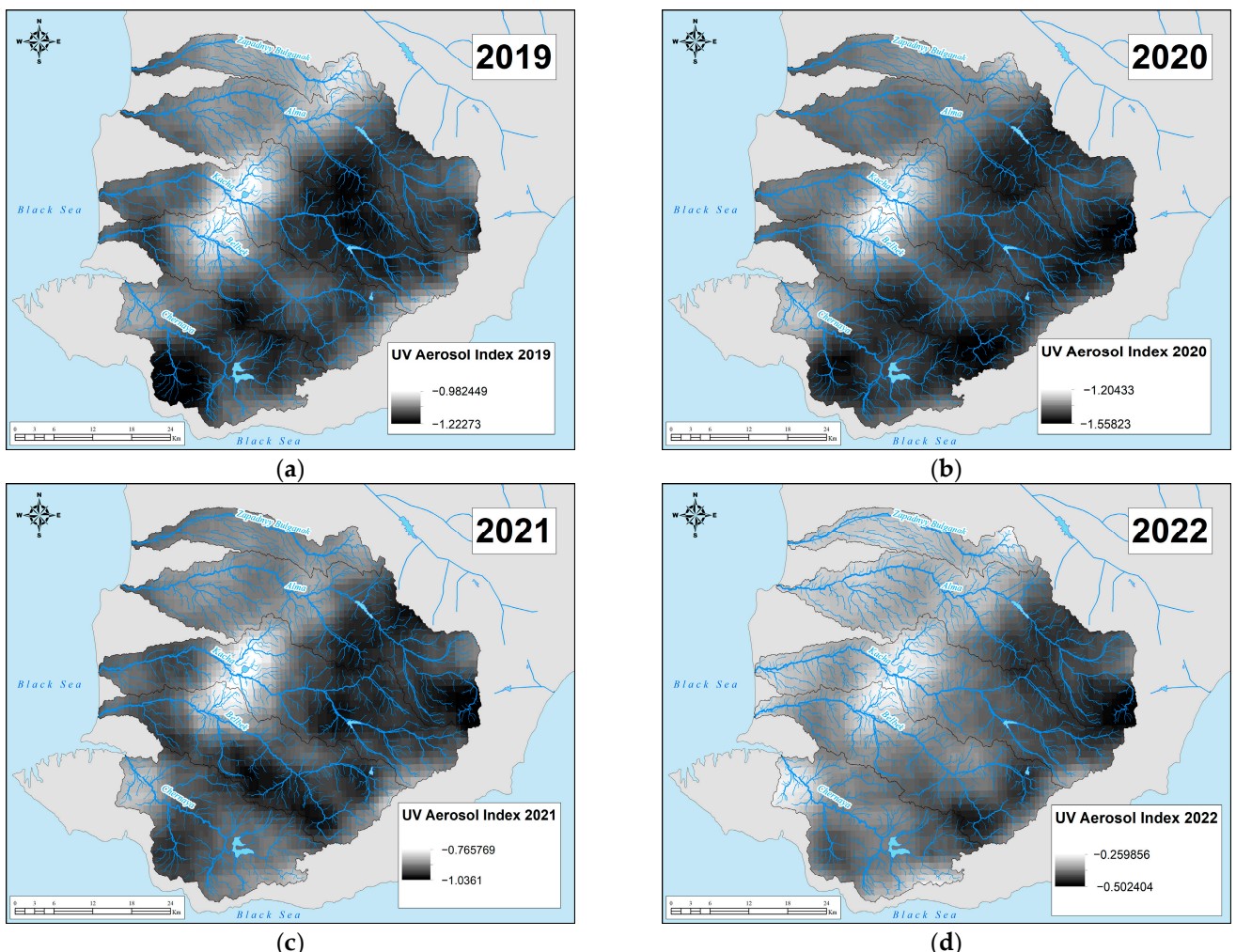

**Figure 8.** Distribution of aerosol content in the atmosphere: (**a**) 2019; (**b**) 2020; (**c**) 2021; (**d**) 2022.

Figure 5 shows the spatial and temporal variability of the field of formaldehyde concentration values within the river basins of the north-western slope of the Crimean Mountains, which reaches maximum values in the southwestern part of the study area (especially in the area of the city of Sevastopol and the surrounding area), while minimum values are observed in the central part.

The decreased concentration of carbon monoxide in the study area from the north-west to the south-east (Figure 6) is associated with the increasing role of the emergence of forest ecosystems in this direction, along with a general decline in population and industrial production. In general, the field of carbon monoxide content values, which has a relatively simple structure, is weakly subject to variability.

Figure 7 depicts the change in methane content from 2019 to 2022. Unfortunately, due to the incompleteness of the dataset for studying the values of the distribution field of methane values in the atmospheric column, it is currently impossible to establish a complete picture for the entire studied territory. However, the available data are sufficient to observe a pattern of increasing methane concentrations from west to east, as well as the formation of localization foci along large settlements.

Figure 8 shows the change in the values of the aerosol index, which registers the presence of aerosol plumes from dust outbreaks, biomass burning, and—in some regions—volcanic ash. The highest values of the aerosol index are achieved in the area of the city of Bakhchysarai, which is associated with the functioning of industrial enterprises comprising sources of emissions in the city and the surrounding area. Although high values are also typical for roads connecting the cities of Sevastopol and Simferopol, as well as the

lower reaches of rivers, while the lowest values are associated with the upper reaches of rivers, local distributions of values are influenced by cities located along the southern coast of Crimea.

In general, the spatial and temporal variability of the sulfur dioxide, carbon dioxide, formaldehyde, carbon monoxide, and methane content in the atmosphere, along with that of the aerosol index, is clearly visible in the Figures 3–8.

Table 1 shows the change in the average annual values of the considered indicators in the context of each river basin. In general, there are slight changes from year to year, including a decline in 2020–2021, and subsequent growth. The reduction in the atmospheric content of various pollutants can be attributed to the influence of strict COVID-19 pandemic restrictions imposed during 2020 and partial restrictions in 2021, as well as the influence of the summation effect and consequent purification of the atmosphere. The increase in the concentration of pollutants may also be due to the 2021 operationalization of the four-lane Tavrida highway, which now significantly affects the state of the atmosphere.

### 3.2. Comparison of Concentration with Background Values within Protected Areas

Since it is impossible to compare the obtained data with the maximum allowable concentration (MAC) figures, a proposed estimation approach uses deviation from the background value characteristic of the protected area territory. In order to assess the deviation from the norm, background values of indicators characteristic of protected areas located in the river basins were used. For this purpose, the average values of the considered indicators from July 2018 to February 2023 were calculated.

The used background values were as follows: nitrogen dioxide—$0.00002$ mol/m$^2$; sulfur dioxide—$0.00018$ mol/m$^2$; carbon monoxide—$0.031$ mol/m$^2$; formaldehyde—$0.0000877$ mol/m$^2$; methane $1850$ mol/mol. The background aerosol index value was $-0.96$. The deviation of the values for each pixel of the image from the background (as a percentage) is shown in Figure 9.

In general, for all indicators (with the exception of sulfur dioxide), the greatest deviations are characteristic of the central part of the study area and the lower reaches of the basins of the rivers under consideration (Figure 9). The complex pattern of the distribution of the sulfur dioxide field, including deviations from the background values, is explained by socio-economic factors—in particular, the use of coal for heating of dwellings and transport from industrial facilities located near Simferopol. Here, the gradual but intensive growth of industrial and agricultural production as compared with 2018 should also be taken into account: this also affects the complex spatial pattern of the concentration of pollutants.

Based on the established background values of the concentration of pollutants above the protected areas, a complex index of atmospheric pollution was calculated (Table 3).

**Table 3.** CIAP within the territory of the river basins of the north-western slope of the Crimean Mountains.

| Year | Study Area as a Whole | River Basin | | | | |
|------|------------------------|------------------|------|-------|--------|----------|
| | | Zapadnyy Bulganak | Alma | Kacha | Belbek | Chernaya |
| 2019 | 5.21 | 5.54 | 5.22 | 5.12 | 5.13 | 5.02 |
| 2020 | 5.40 | 5.64 | 5.50 | 5.49 | 5.22 | 5.12 |
| 2021 | 4.97 | 4.98 | 4.90 | 4.96 | 4.93 | 5.09 |
| 2022 | 5.09 | 5.19 | 5.20 | 5.03 | 4.97 | 5.06 |

As can be seen from Table 3, the lowest values of the complex index of atmospheric pollution within the study area were recorded in 2021; this occurred primarily as a consequence of quarantine policies associated with the COVID-19 epidemic (lockdown, restriction of population movement, lowering of industrial emissions, etc.). When comparing the complex index of atmospheric pollution by river basins, the highest values are characteristic of the Zapadnyy Bulganak River basin, while the lowest values were recorded for the Chernaya River basin.

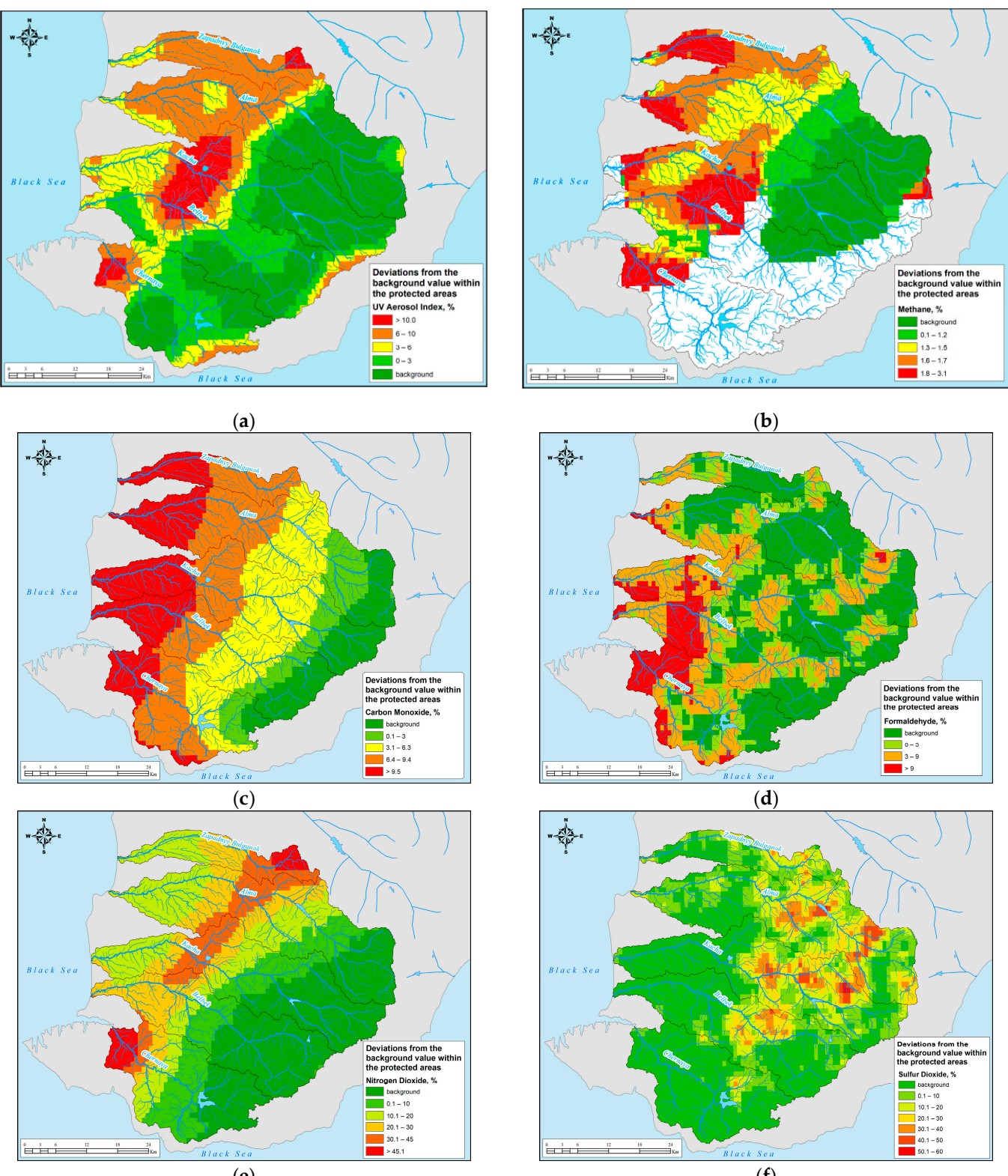

**Figure 9.** Deviations from the background value within the protected areas of the concentration of substances: (**a**) aerosol index; (**b**) methane; (**c**) carbon monoxide; (**d**) formaldehyde; (**e**) nitrogen dioxide; (**f**) sulfur dioxide.

The spatial localization of the complex index of atmospheric pollution from 2019 to 2022 is shown in Figure 10.

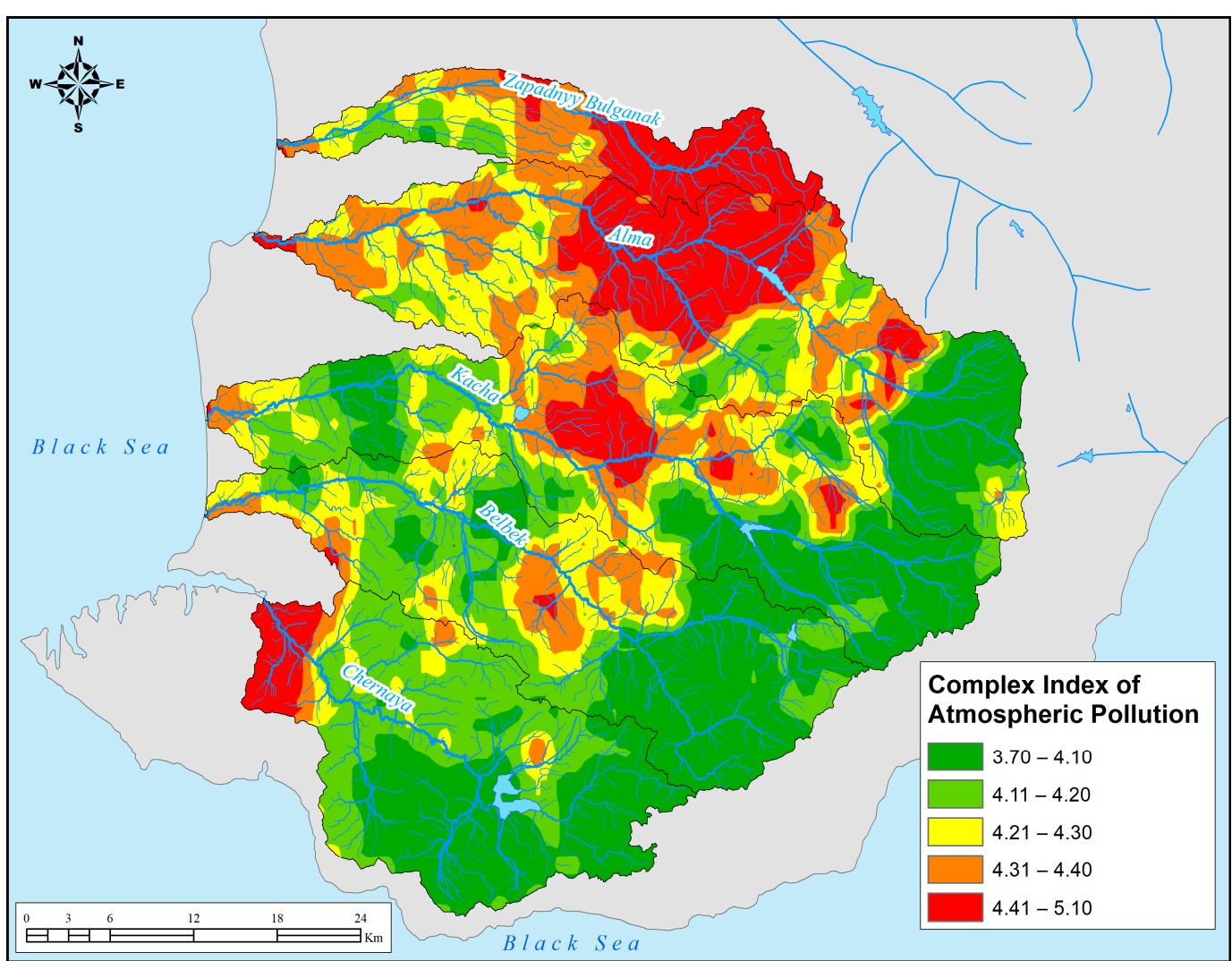

**Figure 10.** CIAP within the territory of the river basins of the north-western slope of the Crimean Mountains for 2019–2022.

Figure 10 shows the main areas of atmospheric pollution within the river basins. As can be seen from Figure 10, the highest values of the complex index of atmospheric pollution are characteristic of large settlements (Simferopol, Bakhchysarai, Sevastopol, Inkerman) and adjacent lands. The minimum values of the complex index of atmospheric pollution within the study area are observed over forests.

### 3.3. Average Monthly Variability of Atmospheric NO$_2$ Concentration

Let us consider the distribution and dynamics of nitrogen dioxide values in the atmosphere within the river basins of the north-western slope of the Crimean Mountains for 2022 in more detail. Figure 11 shows the average monthly values of nitrogen dioxide.

As can be seen from Figure 11, the highest concentrations of nitrogen dioxide values (NO$_2$) occur in winter; they then start to decrease towards the summer before again gradually increasing.

When considering the values of nitrogen dioxide (NO$_2$) content within each catchment area, a similar overall pattern is observed. Figures 12–16 show the average monthly variability of nitrogen dioxide concentration (NO$_2$) within the basins of the Zapadnyy Bulganak, Alma, Kacha, Belbek, and Chernaya rivers.

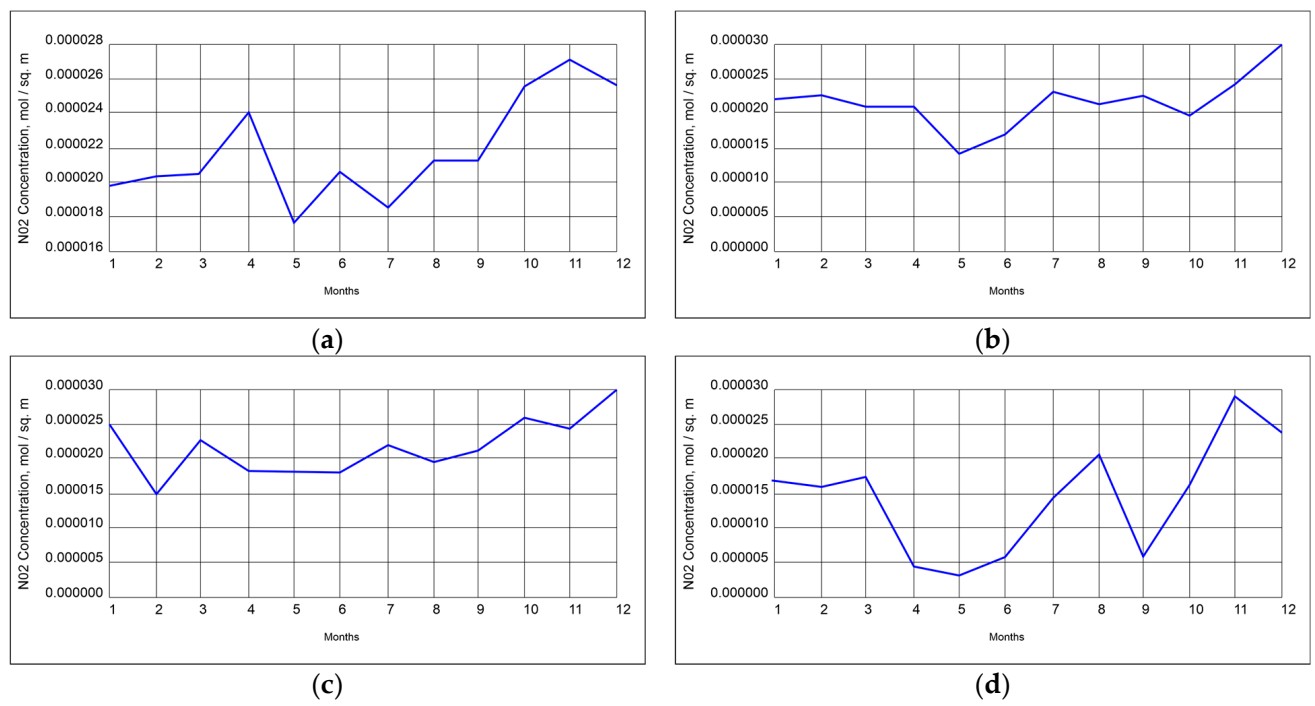

**Figure 11.** Average monthly values of the distribution of nitrogen dioxide (NO$_2$): (**a**) 2019; (**b**) 2020; (**c**) 2021; (**d**) 2022.

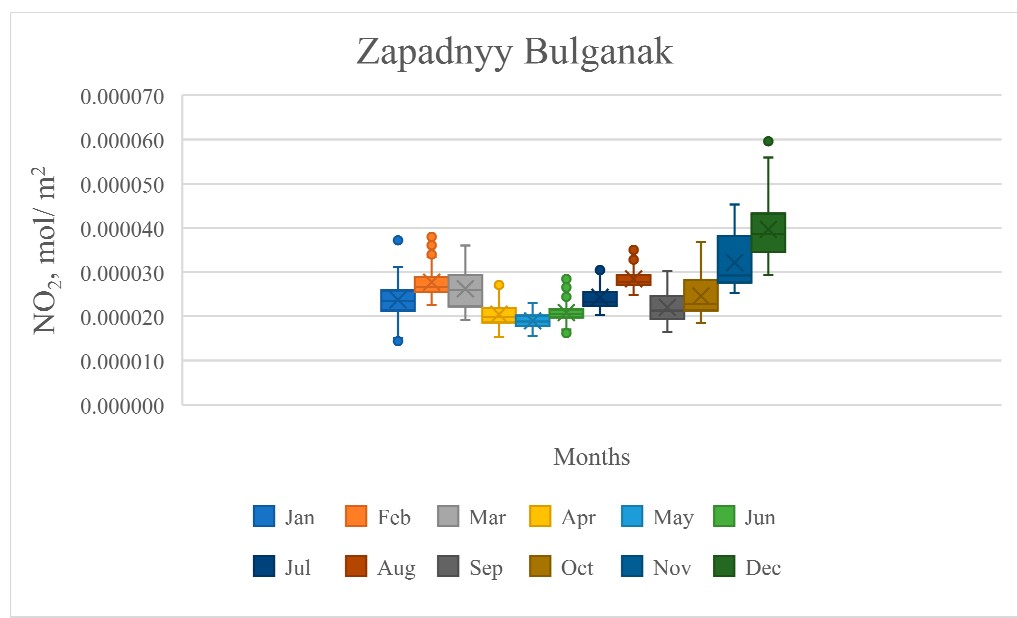

**Figure 12.** Dynamics of atmospheric nitrogen dioxide (NO$_2$) content in the Zapadnyy Bulganak River basin in 2022.

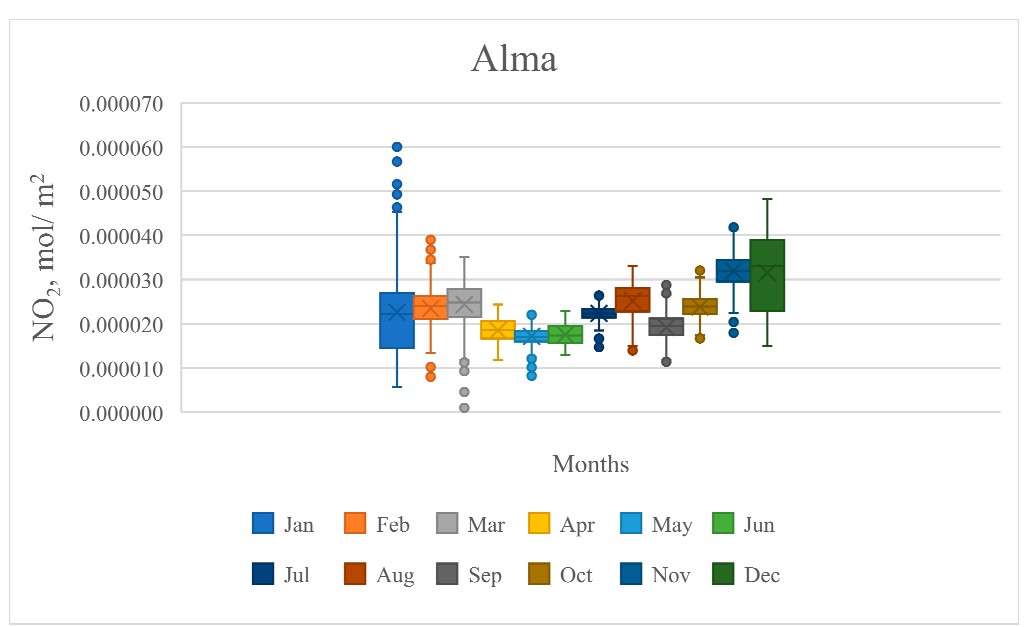

**Figure 13.** Dynamics of atmospheric nitrogen dioxide (NO$_2$) content in the Alma River basin in 2022.

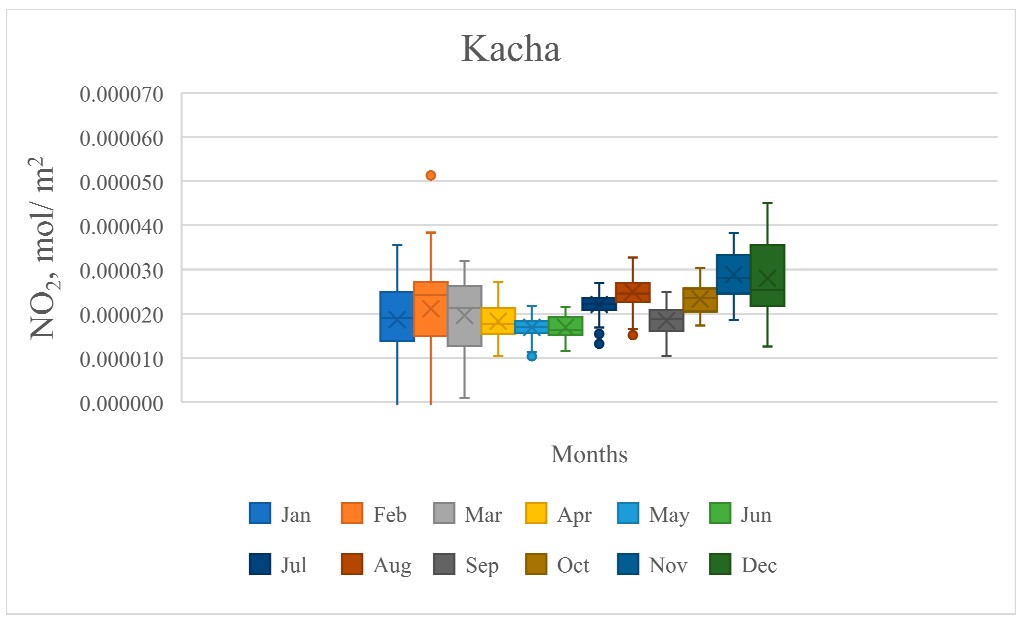

**Figure 14.** Dynamics of atmospheric nitrogen dioxide (NO$_2$) content in the Kacha River basin in 2022.

*3.4. Comparison of Atmospheric Emissions Values Using Ground-Based Monitoring Results and Earth Remote Sensing Data*

To assess the accuracy of the pollutant content values obtained on the basis of Sentinel-5 Precursor satellite data, these were compared with data obtained from monitoring observations [60–63]. For the study area, the data of field monitoring observations are represented by a limited and rather scattered number of points. Monitoring studies are conducted for the cities of Sevastopol, Simferopol, Saki, Alushta, Yalta, and Bakhchysarai, which are either located within the river basins or near them. At the same time, it should be borne in mind that the amounts of pollutant emissions in tons calculated as a result of monitoring for these settlements are not divided into components. In this regard, it becomes problematic to assess the emissions of certain pollutants.

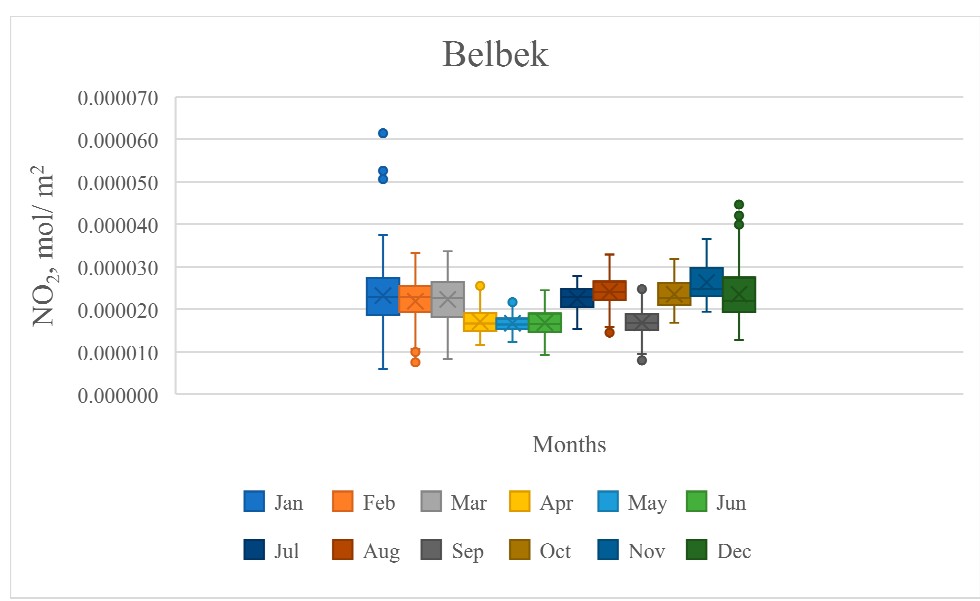

**Figure 15.** Dynamics of atmospheric nitrogen dioxide ($NO_2$) content in the Belbek River basin in 2022.

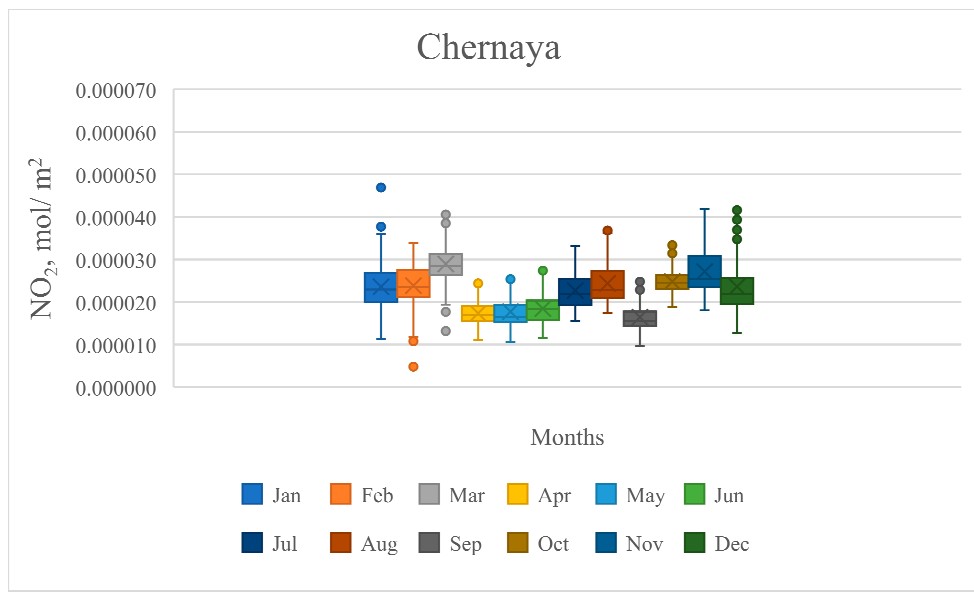

**Figure 16.** Dynamics of atmospheric nitrogen dioxide ($NO_2$) content in the Chernaya River basin in 2022.

To obtain an assessment of quantitative pollution indicators based on monitoring data, it is proposed to use regression analysis and the equation of the relationship of the concentration of substances in the atmosphere with the total amount of pollution within the above-mentioned cities. The relationship equation is obtained based on the use of RStudio (primary data preparation) and Microsoft Excel (data visualization) software.

The field of total emissions of pollutants over the territory of the river basins of the north-western slope of the Crimean Mountains was re-established on the basis of stable connections, as shown in Figure 17.

As can be seen from Figure 17, there is a direct relationship between the quantity of emissions into the atmosphere according to the monitoring results and the concentration of nitrogen dioxide ($NO_2$) having a determination coefficient value of more than 0.85. The obtained coupling equations comprise an important link in determining the total amount of emissions over the catchment area. Obviously, the values recalculated using equations

are much more accurate than the simple interpolation performed on six monitoring points in the cities presented above.

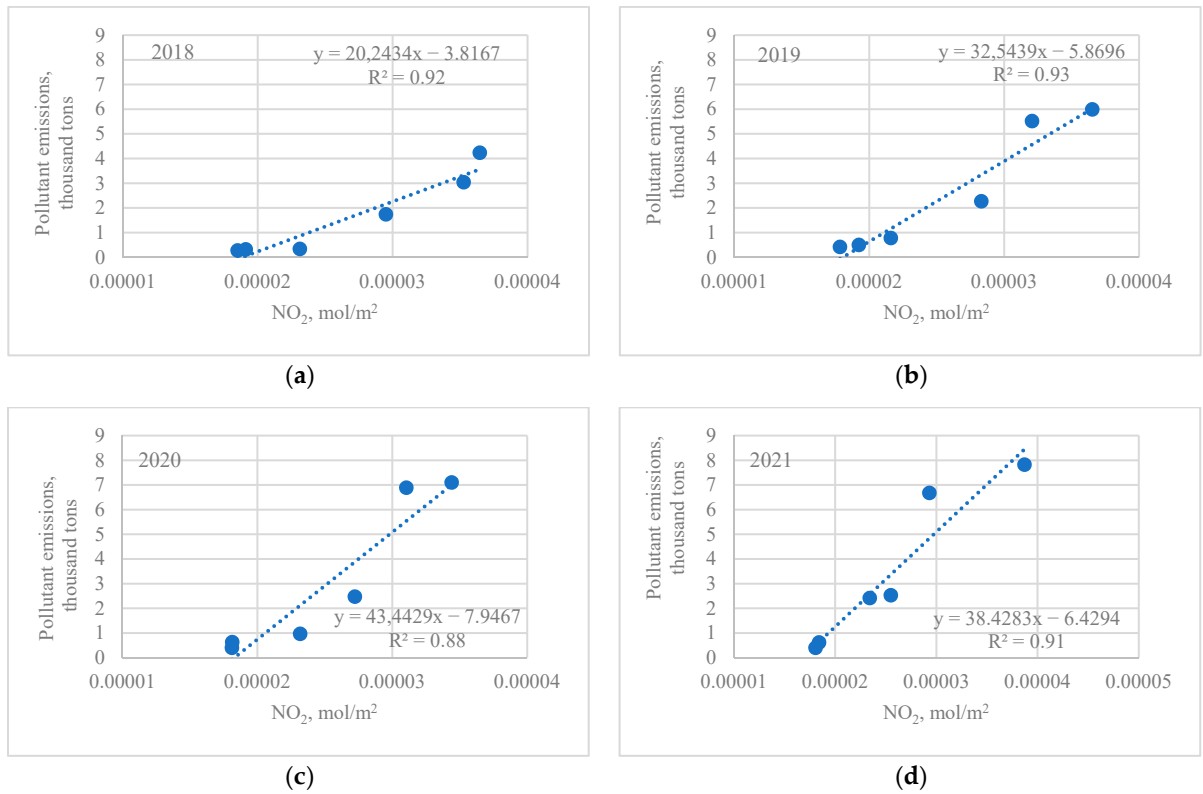

**Figure 17.** The relationship between the concentration of nitrogen dioxide (Sentinel-5 Precursor satellite survey data) and the amount of pollutant emissions (monitoring data) for the cities of Sevastopol, Simferopol, Saki, Alushta, Yalta, and Bakhchysarai: (**a**) 2018; (**b**) 2019; (**c**) 2020; (**d**) 2021.

Figure 18 shows maps of the emission fields of atmospheric pollutants within the river basins. As can be seen from Figure 18, an increase in emissions is observed from 2018 to 2021, while in 2018, there were three main centers of pollutants into the atmosphere, comprising Simferopol, Bakhchysarai, and Sevastopol, the impact of whose emissions expanded in 2021 along the Tavrida Highway connecting them.

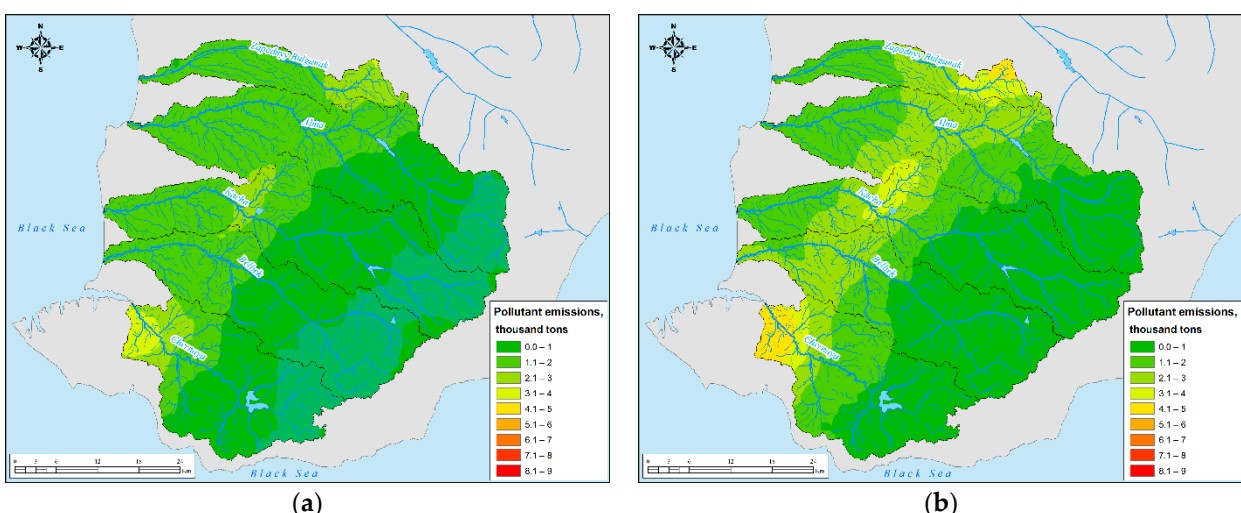

**Figure 18.** *Cont.*

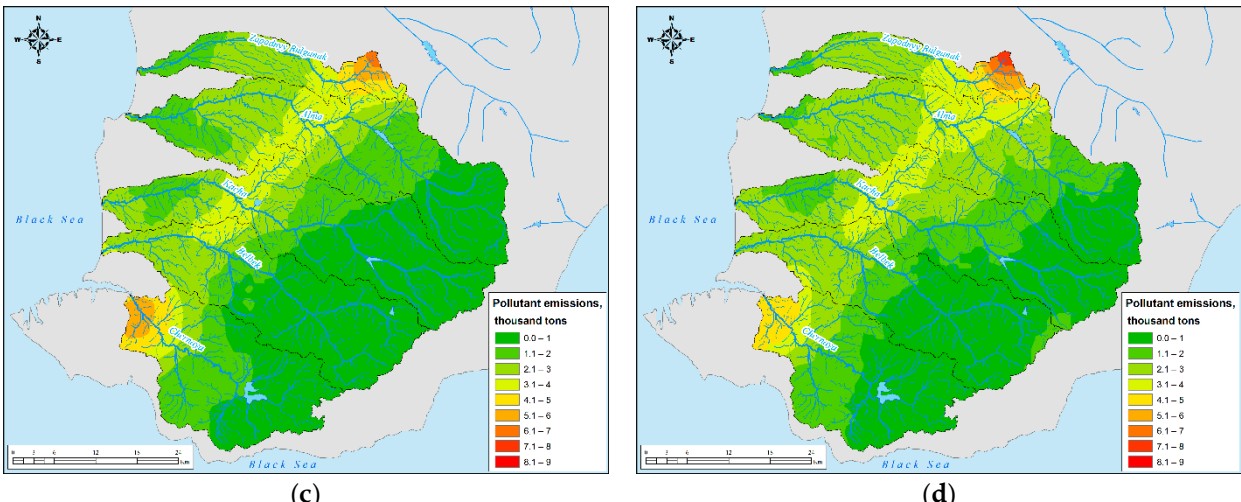

**Figure 18.** Quantity of pollutant emissions (thousand tons): (**a**) 2018; (**b**) 2019; (**c**) 2020; (**d**) 2021.

The least affected by pollution is the south-eastern part of the study area, where emissions, in most cases, are absent or minimal; however, the area of such territories decreased from 2018 to 2021.

## 4. Discussion

At the present stage of the development of scientific knowledge, Sentinel-5 Precursor data is a valuable resource for obtaining information about atmospheric pollution by various substances. The use of Sentinel-5 Precursor data in combination with a geographic information system (ArcGIS) and remote cloud computing platform (GEE) forms the basis for large-scale studies of the dynamics of pollutant emissions within river catchments, which represent the most valuable sources of high-quality fresh water. The key advantage of using GEE for Sentinel-5 Precursor data analysis is the powerful and flexible platform it provides for processing large-scale geospatial data and performing complex analysis tasks. Among the main advantages of the Sentinel-5 Precursor satellite is the free availability of data, which can be used for various research purposes; moreover, the almost daily updated data have a global coverage, allowing the studied values to be compared with those obtained from different regions of the world. Despite its many advantages, some disadvantages of the Sentinel-5 Precursor system should also be considered, including the limitation on the number of measured parameters of gases and aerosols, the inability to assess the state of the atmosphere in every part of the world in online regime, complex data acquisition and processing, as well as the need to use special software (reading netCDF files). While the spatial resolution of the received data is not high (1113.2 m or 0.01 degrees), it is still the best among currently available devices for monitoring the state of the atmosphere. Nevertheless, Sentinel-5 Precursor satellite data cannot be used to assess the total impact of the quantity of emissions of various substances into the atmosphere. For these purposes, only statistical information and data obtained from field research and monitoring can be effectively used.

Various research methods are used to evaluate the substance content of the tropospheric column according to the Sentinel-5 Precursor satellite data and calculations based on field observations. It is obvious that satellite images are founded on a more rigorous scientific basis, while monitoring data is limited due to the time of information collection and the amount of information provided on a spatiotemporal basis. However, the satellite survey data can, in turn, be used to recalculate the data on the total amount of emissions into the atmosphere in quantitative terms. While the obtained dependence equations show high values of this relationship, the most significant relationship is that characterizing the total amount of pollutant emissions and the concentration of carbon dioxide in the troposphere. To overcome limitations in conducting future research, it is necessary to signif-

icantly improve the in situ system of monitoring and sampling air pollutants. It is evident that the network of sampling stations should encompass a larger number of observation points and include a wider temporal interval of study, including monthly and seasonal breakdowns, rather than solely relying on annual values. A significant improvement in future work would involve enhancing the resolution of air pollution data provided by the Sentinel satellite; this would allow for the identification not only of general regional patterns, but also of the unique local characteristics of air pollution caused by various chemical substances.

In addition to functioning as an indicator of the content of nitrogen monoxide (NO), the data on the atmospheric $NO_2$ content was used to re-establish the spatial field of emission values within the river basins. Nevertheless, compared with nitrogen dioxide, other indicators of substance concentrations considered in the work (formaldehyde, methane, sulfur dioxide, nitrogen oxide) did not show high values of $R^2$; for this reason, the restoration of the emission field was carried out using the nitrogen dioxide concentration values. This is justified by the fact that nitrogen dioxide and nitrogen oxide mainly enter the atmosphere due to the results of anthropogenic human activity (burning of fossil fuels and biomass), and only to a lesser extent as a result of natural processes (microbiological processes in soils, forest fires). Thus, given the insufficient spatiotemporal series of observations, the data from the Sentinel-5 Precursor space satellite can be used to restore the atmospheric pollutant concentration values.

The decrease in the atmospheric content of various pollutants in 2020–2021 is global in nature due to the worldwide impact of the COVID-19 pandemic. As emphasized by many researchers, this observed phenomenon is also typical of other regions of the world [64–67]. However, within the river basins under consideration, which have a relatively small area, the drop in emissions is not as pronounced as in many other countries [64–66,68–71]. When comparing the values obtained in our study for the year 2019 with the results obtained in [26] for the territory of North Macedonia, it can be stated that the $NO_2$ concentration values for the largest cities in North Macedonia (Skopje, Bitola, Prilep) exceed the values in the largest inhabited areas within the investigated territory by 2–3 times. Comparing them to the $NO_2$ concentration values in Turkey [25], the concentration excess in the Istanbul region is 40 times greater than within the major inhabited areas within the river basins of the north-western slopes of the Crimean Mountains.

The concentration of CO in major inhabited areas within the river basins of the north-western slopes of the Crimean Mountains reaches 80–90% of the maximum values within North Macedonia [26]. When comparing them with the largest cities in Iran (Tehran, Isfahan, Tabriz) [69], it can be stated that the $NO_2$ concentration in the air within major inhabited areas within the river basins of the north-western slopes of the Crimean Mountains is more than 100 times lower. However, it is important to consider the factor that Bachisaray (one of the largest cities within the study area) has a population of approximately 30,000 people, while Tehran has around 9 million inhabitants and Isfahan has 2 million inhabitants. Overall, taking into account the predominance of forests and protected natural areas within the river basins of the north-western slopes of the Crimean Mountains, atmospheric pollution is established to be significantly lower compared to other regions of the world where similar studies have been conducted.

The high content of atmospheric pollutants in the Zapadnyy Bulganak River basin as compared with other studied basins is primarily due to its minimal afforestation, resulting in a failure to capture part of the emissions, as well as to the significant influence of the city of Simferopol, located next to the basin. The comparatively lower air pollution in the Alma, Kacha, Belbek, and Chernaya river basins is primarily due to the large forested areas located in their upper reaches.

A significant limiting factor of this study consists in the lack of officially published statistics on atmospheric air pollution. For example, Figures 16 and 17 do not show the results for 2022, since officially published data on the number of emissions in the cities of

Sevastopol, Simferopol, Bakhchysarai, Yalta, Alushta, and Saki had not been published at the time of writing, making it impossible to conduct a regression analysis.

A large number of works have been aimed at evaluating the accuracy of the ESA/EU Copernicus Sentinel-5 Precursor dataset [21,22,72–75]. The work [76] indicates that the accuracy of Sentinel-5 data in clean or slightly polluted conditions averages −23 to −37%, reaching −51% in heavily polluted areas. In [77], the accuracy of the CO, $NO_2$, and $SO_2$ air quality parameters obtained from the Sentinel-5 precursor data sets was found to be 89.5%, 83.54%, and 86%, respectively, as compared with on-site measurements. In [78], it is indicated that accuracy can reach more than 1.5%. Our research demonstrates a high correlation of measurements for the concentration of nitrogen dioxide in cities with the data obtained via satellite images.

As well as playing a crucial role in understanding the spatial localization of pollution sources and the consequences of air pollution, Sentinel-5 Precursor satellite images can be used to develop effective strategies to reduce the impact of such emissions. In order to minimize the effects of atmospheric pollution, it is necessary to implement a comprehensive policy aimed at reducing emissions from the main sources of pollution. Strategies that can be considered as part of a comprehensive policy to combat air pollution include the following: implement a policy of introducing renewable energy sources, such as solar energy, wind energy, and hydropower, which will contribute to reducing emissions into the atmosphere; constantly improve and introduce advanced technologies and equipment in all areas of industry; implement strict emission standards and regulations for industry and transport. In the field of transport, it is also important to switch from internal combustion engines to electric motors and other types of engines that reduce emissions. Planting trees and creating green spaces can help to reduce air pollution by absorbing carbon dioxide and other pollutants while improving air quality. The inclusion of green areas in urban planning can help to reduce the concentration of pollutants in densely populated areas. Raising public awareness of the health and environmental impacts of air pollution is important. Educational campaigns can inform people about the sources and consequences of air pollution, as well as promote sustainable behaviors and lifestyle choices that contribute to air purification.

## 5. Conclusions

The present paper presents the results of modeling the content of pollutants—nitrogen dioxide, sulfur dioxide, carbon monoxide, formaldehyde, and methane, as well as the aerosol index—within the basins of the Zapadnyy Bulganak, Alma, Kacha, Belbek, and Chernaya rivers. In addition to analyzing the concentrations of various pollutants in the atmosphere, this study reveals their spatial and temporal distribution within specific watersheds. This study revealed that, in most cases, the highest concentrations of pollutants were observed in areas of river basins that encompass major urban centers, particularly Bachisaray, Sevastopol, and Inkerman, as well as along major highways. Additionally, due to the transport of air masses, the nearby city of Simferopol has a significant influence on the studied river basins. The obtained results for the river basins on the north-west slope of the Crimean Mountains are significant due to the lack of permanent air pollution monitoring centers in the region, which is compounded by the significant growth of transportation and industrial activities. Remote sensing research methods play a crucial role in addressing this challenge by providing valuable insights into the atmospheric pollution patterns and trends. This can be useful for establishing the release points of pollution sources and developing measures to reduce their impact. The presented findings can be useful not only for environmentalists and climate scientists, but also for government authorities and industrial companies. The authors hope that this can form a starting point for developing more effective measures to reduce emissions of pollutants in order to improve the environmental situation in the region. Among the prospective directions for future research involving the use of atmospheric pollution data with the application of Sentinel-5 satellite imagery, several potential avenues can be highlighted. Within the

Crimean Peninsula, it is crucial to significantly expand the utilization of Sentinel-5 data for other river basins, since the river basins on the north-west slope of the Crimean Mountains are not the most polluted areas. Making comparisons of pollution levels between different river basins would be particularly interesting. Of specific interest is the examination of atmospheric pollution in the basin of the largest river on the Crimean Peninsula—the Salgir River. Studies on the accumulation of pollutants by vegetation (in leaves and branches of trees, in mosses, in lichens), along with the establishment of a relationship between the quantity of pollutants in the atmosphere and their absorption/accumulation, may also be promising. It goes without saying that it is necessary to continue research and analyze Sentinel-5 satellite images in order to identify and track sources of emissions in river basins (and other research objects), such as industrial enterprises, power plants, road transport, etc. An important factor in these studies involves not only the study of local peaks of pollutant emissions, but also an investigation into changes in the quantity of emissions over time—that is, spatiotemporal dynamics. Additionally, it is essential to acknowledge the constraints associated with utilizing data from the Sentinel-5 satellite imagery. The comprehensiveness and spatial coverage of Sentinel-5 data should be taken into account, given that these data have been made available since 2018, and their extent of coverage may vary across different regions worldwide. Another noteworthy limitation pertains to the necessity of conducting preprocessing and data correction procedures. Moreover, a substantial limitation of this study lies in the restricted number of chemical compounds under observation. It is imperative to expand the repertoire of substances monitored in future Sentinel research missions as part of the ongoing research program.

**Author Contributions:** Conceptualization, V.T., R.G. and T.G.; methodology, V.T. and R.G.; software, V.T.; validation, V.T. and T.G.; formal analysis, V.T.; investigation, V.T., R.G. and T.G.; resources, V.T.; data curation, V.T.; writing—original draft preparation, V.T.; writing—review and editing, R.G. and T.G.; visualization, V.T.; supervision, R.G.; project administration, V.T. All authors have read and agreed to the published version of the manuscript.

**Funding:** The research was conducted within the framework of the research topic of IBSS, registration number: 121040100327-3. The RUDN University Strategic Academic Leadership Program supported this research.

**Data Availability Statement:** The data presented in this study are available on request from the corresponding author.

**Conflicts of Interest:** The authors declare no conflict of interest. The funders had no role in the design of this study; in the collection, analysis, or interpretation of data; in the writing of the manuscript; or in the decision to publish the results.

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
