# Peer review of "Unveiling Air Pollution in Crimean Mountain Rivers: Analysis of Sentinel-5 Satellite Images Using Google Earth Engine (GEE)"

_remotesensing, doi:10.3390/rs15133364_

Round 1

Reviewer 1 Report

By using the Sentinel-5 satellite images, this manuscript analysis the atmospheric air pollution in Crimean, the resulst is intresting. but the manuscript should make a minor revision before publication

1. the title shoud revisd : The title Sentinel-5 satellite images for the analysis of atmospheric air pollution.

2. abstract should add study method.

3.There are too many references quoted in the introduction, so it needs to be simplified and keep those related to this study.

 Minor editing of English language required

Author Response

Response to Reviewer 1 Comments

Dear reviewer.

Thank you for conducting the review of my scientific article. I appreciate your thorough evaluation and valuable feedback.

Point 1: the title shoud revisd : The title Sentinel-5 satellite images for the analysis of atmospheric air pollution.

Response 1: Thank you for your feedback. I appreciate your suggestion (which was also mentioned by Reviewer 2) regarding the title of the article, and I have made the necessary changes accordingly.

Point 2: abstract should add study method.

Response 2: Based on your suggestion, we have included a section in the abstract of the scientific article outlining the research methods employed. This addition provides a clearer understanding of the approach and methodology used in the study.

Point 3: There are too many references quoted in the introduction, so it needs to be simplified and keep those related to this study

Response 3: I apologize for any inconvenience caused by the extensive literature review in the introduction. I understand your concern about its relevance to the research problem. In response to your feedback and the request of the chief editor, I will carefully reconsider the literature review section. I will focus on including only the most pertinent and directly related sources to ensure the coherence and conciseness of the article. Additionally, I will incorporate the additional sources suggested by the chief editor to enhance the comprehensiveness of the review. I appreciate your valuable input and will make the necessary revisions to address these concerns appropriately.

Reviewer 2 Report

The presented  article with title Sentinel-5 satellite images for the analysis of atmospheric air 2 pollution using remote research methods and the Google Earth 3 Engine (GEE) cloud computing platform 

The objectives of the work are: 

(1) to show the spatiotemporal variability of pollutant concentration fields within the river basins of the north-western slope of the Crimean Mountains calculated from Sentinel-5 Precursor satellite images; 

(2) to establish a link between the concentration of pollutants from satellite images and the results  of monitoring atmospheric pollution (for individual settlements); 

(3) on the basis of satellite imagery data, to reconstruct the concentration field of pollutant emissions.

Introduction, Materials and Methods, results, and Discussion presented very well.

In general the topic is sounded and within the journal topic. There fore it is sutabilr for the Special Issue Google Earth Engine for Remote Sensing Big Data Landscapes and the Section, Remote Sensing Image Processing.

minor comment 

1-title is too long, please short it in informative view (suggested)

Analysis of atmospheric air pollution using remote research methods and the cloud computing platform

2-aims or objects should present as a paragraph not as points.

3-Figures 10 and 16, Please increase the font of axis information to be clear.

4-Conclusion is very short please represent all finding with conclusion to be more informative 

Author Response

Dear reviewer.

Thank you for conducting the review of my scientific article. I appreciate your thorough evaluation and valuable feedback.

1-title is too long, please short it in informative view (suggested)

Analysis of atmospheric air pollution using remote research methods and the cloud computing platform

Response 1: Thank you for your feedback. I appreciate your suggestion (which was also mentioned by Reviewer 1) regarding the title of the article, and I have made the necessary changes accordingly.

2-aims or objects should present as a paragraph not as points.

Response 2: Done

3-Figures 10 and 16, Please increase the font of axis information to be clear.

Response 3: Done

4-Conclusion is very short please represent all finding with conclusion to be more informative 

Response 4: Done

Reviewer 3 Report

Abstract needs to be re-written. It should include briefly all elements of the paper.

Study location description needs to be corrected. It is described in to terms: once as north-western slope 115 of the Crimean Mountains and second as southwestern part of the Crimean Peninsula which is supported by Figure 1.

Chat figures need to be presented in a clear professional format rather than copy/paste from excel sheet.

Methods section:

It is too short.

Provide more technical data on the satellite used. 

What type of ArcGIS software was used?

Provide more details about it.

Provide more details about the GEE algorithms used in this study.

Table 1: should move to the results section.

Table 2: use (.) instead of (,) for decimal points

Table 2: include chemical abbreviations with no description in the manuscript

Figure 7: legends not clear

Figure 11-15: what is x–axis

Figure 16: Y and X axis’s

Results and Discussion sections is not covering all the targeted parameters.

Moderate editing of English language required

Author Response

Response to Reviewer 1 Comments

Dear reviewer.

Thank you for conducting the review of my scientific article. I appreciate your thorough evaluation and valuable feedback.

Point 1: Abstract needs to be re-written. It should include briefly all elements of the paper..

Response 1: The abstract of the article has been corrected.

Point 2:. Study location description needs to be corrected. It is described in to terms: once as north-western slope 115 of the Crimean Mountains and second as southwestern part of the Crimean Peninsula which is supported by Figure 1

Response 2: Done

Point 3: Chat figures need to be presented in a clear professional format rather than copy/paste from excel sheet

Response 3:. Dear Reviewer. I didn't quite understand your remark. Previously, when submitting articles to MDPI journals, there were no difficulties using graphs presented in excel.

Point 4:. Methods section: It is too short

Response 4: We have expanded section 2.

Point 5: Provide more technical data on the satellite used

Response 5:. Done

Point 6:. What type of ArcGIS software was used? Provide more details about it

Response 6: Done. ArcGIS 10.8

Point 7: Provide more details about the GEE algorithms used in this study

Response 7:. We included in the revised version of the article the algorithm for studying air pollution in river basins using GIS and GEE (Figure 2).

Point 8:. Table 1: should move to the results section

Response 8: Table 1 is not the results of a study. It represents statistics (monitoring) data with which the data obtained as a result of calculations are further compared. Therefore, it should not be moved to Section 3 Results.

Point 9: Table 2: use (.) instead of (,) for decimal points Table 2: include chemical abbreviations with no description in the manuscript

Response 9:.Done

Point 10:. Figure 7: legends not clear

Response 10: Done We increased the size of the legend to the geographic map. Text is now clearer and easier to read

Point 11: Figure 11-15: what is x

Response 11:. Done

Point 12:. Figure 11-15: what is x

Response 12: Done

Point 13: Figure 16: Y and X axis’s

Response 13:. Done

Point 14: Results and Discussion sections is not covering all the targeted parameters

Response 14:. The Discussion section has been updated with new data. The Results section contains all the data we received

Round 2

Reviewer 3 Report

Authors sufficiently incorporated the reviewer’s comments. Some figures needs to be replaced with more clear one such as figure 2, figure 9 and figure 18 .

No Comments

Author Response

Dear reviewer. We are grateful for your work. We have improved the quality of the drawings that you kindly pointed out to us.